# Current and Future Role of Neoadjuvant Chemoimmunotherapy for Early Triple-Negative Breast Cancer: Which Way to Go Forward

**DOI:** 10.3390/medicina58050600

**Published:** 2022-04-27

**Authors:** Alessandro Rizzo, Antonio Cusmai, Gennaro Gadaleta-Caldarola, Gennaro Palmiotti

**Affiliations:** 1Struttura Semplice Dipartimentale di Oncologia Medica per la Presa in Carico Globale del Paziente Oncologico “Don Tonino Bello”, I.R.C.C.S. Istituto Tumori “Giovanni Paolo II”, Viale Orazio Flacco 65, 70124 Bari, Italy; antoniocusmai@hotmail.com (A.C.); gennaropalmiotti@hotmail.it (G.P.); 2Medical Oncology Unit, ‘Mons. R. Dimiccoli’ Hospital, Azienda Sanitaria Locale Barletta, 76121 Barletta, Italy; gergad@libero.it

**Keywords:** breast cancer, immunotherapy, immune checkpoint inhibitors, pembrolizumab, atezolizumab

## Abstract

Immunotherapy has revolutionized previous triple-negative breast cancer (TNBC) treatment algorithms, prompting researchers and clinicians to consider the expansion of the role of immunotherapy in other settings, including the earlier stage of the disease (e.g., as neoadjuvant and adjuvant therapy). The role of chemoimmunotherapy have been assessed in some recently presented and published clinical trials, including the KEYNOTE-522, the IMpassion031, and the GeparNUEVO. In the current Editorial, we will provide a critical snapshot of these studies, exploring strengths and limitations of neoadjuvant chemotherapy in early TNBC.

## 1. Introduction

The last decade has witnessed the emergence of modern immunotherapy, with immune checkpoint inhibitors (ICIs), administered as monotherapy or in combination with other anticancer agents, making a breakthrough in several hematological and solid tumors, including non-small cell lung cancer (NSCLC), hepatocellular carcinoma (HCC), melanoma, renal cell carcinoma, and bladder cancer [1,2,3,4]. ICIs boost cytotoxicity of T cells and block down-regulators of immunity including programmed cell death protein 1 (PD-1), programmed death ligand 1 (PD-L1), cytotoxic T-lymphocyte antigen 4 (CTLA-4), and lymphocyte-activation gene 3 (LAG-3) [5]. ICIs have been also recently explored in breast cancer. Firstly, monotherapy with ICIs has reported disappointing results in metastatic triple-negative breast cancer (TNBC) [6,7,8]. In particular, the KEYNOTE-086 and the KEYNOTE-119 trials evaluating pembrolizumab monotherapy observed low response rates, and these results have been corroborated by other clinical studies evaluating single-agent atezolizumab [9]. Thus, combination strategies have been investigated, given the synergistic effect of ICIs plus other anticancer agents. Among these, chemoimmunotherapy has entered everyday clinical practice as new first-line therapy in metastatic TNBC patients with PD-L1 overexpression (tumours with ≥1% PD-L1 expression in immune cells) or an elevated Combined Positive Score (CPS) [10]. Moreover, several phase I to III clinical trials have been designed to assess immune-based combinations, with these studies having the potential to further modify the therapeutic algorithm of this patient population [11,12]. Based on these premises, ICIs have revolutionized previous TNBC treatment algorithms, prompting researchers and clinicians to consider the expansion of the role of immunotherapy in other settings, including the earlier stage of the disease (e.g., as neoadjuvant and adjuvant therapy) [13,14]. The role of chemoimmunotherapy have been assessed in some recently presented and published clinical trials, including the KEYNOTE-522, the IMpassion031, and the GeparNUEVO.

In the current Editorial, we will provide a critical snapshot of these studies, exploring strengths and limitations of neoadjuvant chemotherapy in early triple-negative breast cancer.

## 2. KEYNOTE-522

The phase III KEYNOTE-522 trial assigned TNBC patients to carboplatin—paclitaxel (4 cycles), followed by 4 additional cycles of anthracyclines-based chemotherapy, plus the PD-1 inhibitor pembrolizumab versus the same pre-operative chemotherapy plus placebo [15]. Neoadjuvant therapy was followed by surgery and postoperative pembrolizumab or placebo for one year of treatment; pathological complete response (pCR) and event-free survival (EFS) were the coprimary endpoints of this phase III trial in the intention-to-treat (ITT) population. According to baseline characteristics of patients, more than 80% of subjects presented PD-L1 positive TNBC, based on the 22C3 assay, and approximately 75% were T1/T2. KEYNOTE-522 met both its primary endpoints, and after a median follow-up of 38.5 months, a statistically and clinically significant EFS advantage was highlighted in the experimental arm, with a Hazard Ratio (HR) of 0.63 (95% CI, 0.48–0.82; *p* = 0.00031) for a 3-year EFS of 84.5% and 76.8% in the experimental and the control arms, respectively [16]. At the first interim analysis, the percentage of TNBC patients with pCR was 64.8% in the pembrolizumab–chemotherapy group and 51.2% in the placebo–chemotherapy group (*p* < 0.001). With mature overall survival (OS) results pending, a slight trend favoring pembrolizumab was observed; in addition, pCR was significantly higher in TNBC patients receiving pembrolizumab plus neoadjuvant chemotherapy than among those receiving placebo—chemotherapy [17]. Of note, TNBC patients showing pCR had optimal outcomes regardless of the use of immunotherapy (3-year EFS of 94.4% and 92.5% in the experimental and the control arm, respectively), while the addition of immunotherapy provided a 3-year EFS benefit compared to placebo alone (67.4% and 56.8%, respectively) in patients with residual disease. As regards the incidence of treatment-related adverse events of grade 3 or higher, this was 78% in the pembrolizumab–chemotherapy group and 73.0% in the placebo–chemotherapy group, including death in 0.4% (3 patients) and 0.3% (1 patient), respectively. In summary, KEYNOTE-522 clearly met its primary endpoints (pCR and EFS) and, with OS data still immature, a trend favoring pembrolizumab was described. Based on these results, the Food and Drug Administration (FDA) approved pembrolizumab as part of neoadjuvant treatment for high-risk, non-metastatic TNBC.

## 3. IMpassion031

The IMpassion031 phase III trial compared neoadjuvant atezolizumab versus placebo combined with sequential nab-paclitaxel (12 weeks) and doxorubicin-cyclophosphamide (4 cycles) for treatment of early stage TNBC [18]. In this randomized, double-blind study, TNBC patients were stratified according to disease stage (II versus III) and PD-L1 status; pCR was 57.6% (95/165) in the immunotherapy-containing arm and 41.1% (69/168) in the chemotherapy alone group, with an absolute difference of 16.5% favoring the addiction of atezolizumab to neoadjuvant treatment [19].

In addition, EFS, disease-free survival (DFS), and OS trends showed a benefit for atezolizumab, without reaching the statistical significance and with data still immature. The incidence of all grade and grade 3–4 treatment-related adverse events, as well as the discontinuation rate, were similar between the two groups; nonetheless, treatment-related serious adverse events were more frequent in the experimental arm. The most frequent all-grade adverse events in the neoadjuvant phase were chemotherapy-related, including alopecia, nausea, diarrhea, and anemia, while the most reported grade 3–4 events were hematological.

## 4. GeparNUEVO

In the phase II GeparNUEVO study, the investigators assigned 174 patients to a sequential treatment with durvalumab, nab-paclitaxel plus durvalumab, and anthracyclines plus durvalumab or to placebo, nab-paclitaxel plus placebo, and anthracyclines plus placebo, followed by surgery [20]; pCR was the primary outcome of this trial, with invasive disease-free survival (iDFS), distant disease-free survival (DDFS), and OS investigated as secondary outcomes. According to the recently presented results of this study, GeparNUEVO failed to meet its primary endpoint, highlighting a non-statistically significant pCR advantage in the durvalumab arm (53.4% and 44.2% in the experimental and the control group, respectively) (Odds Ratio 1.45; 95% CI, 0.80–2.63). However, secondary endpoints such as iDFS, DDFS, and OS showed a statistically significant improvement in patients receiving neoadjuvant durvalumab [20]. In particular, 3-year OS was 95.2% and 83.5% in the chemoimmunotherapy arm and the chemotherapy alone arm, respectively, with a HR of 0.24 (95% CI, 0.08–0.72). As regards treatment-related adverse events, the most common immune-related toxicities were thyroid dysfunction of any grade, which was reported in 47% of cases.

## 5. What the Future Will Be Like: Open Questions

If single-agent ICIs reported an overall limited activity in TNBC patients, combinatorial strategies with cytotoxic chemotherapy have highlighted practice-changing results in early and metastatic disease. In fact, immunotherapy has become an important tool in the therapeutic armamentarium for TNBC, with several PD-1/PD-L1 inhibitors demonstrating a clinical benefit in this setting. As regards neoadjuvant treatment, three recent trials have reported improved clinical outcomes with the addition of atezolizumab, pembrolizumab, and durvalumab: KEYNOTE-522, IMpassion031, and GeparNUEVO, respectively. The aggressive nature of TNBC and its propensity to recur makes this group of breast cancers a very challenging one; positive signals emanating from neoadjuvant chemoimmunotherapy should encourage the scientific community to persist in the long road toward finding more effective treatments for early TNBC.

In a setting with several unanswered issues, fundamental questions come to mind and represent crucial current and future challenges. Among these, some high-risk patients do not benefit from the addition of ICIs to the neoadjuvant chemotherapy backbone and could be spared from unnecessary toxicity. In particular, the identification of this patient population remains mandatory, and this point is particularly important also in terms of costs since the addition of immunotherapy as part of the neoadjuvant approach is a considerable additional expense and will soon represent an obstacle for several countries. For example, T1N0 patients were not included in KEYNOTE-522, and neoadjuvant pembrolizumab should be avoided in this subgroup; similarly, as previously reported, the same trial reported little benefit of immunotherapy in patients achieving pCR, who performed well regardless of pembrolizumab (3-year EFS of 92.5% in the placebo arm).

Secondly, available evidence suggests that in early TNBC, PD-L1 expression by immunohistochemistry is not able to identify those patients that do not benefit from the addition of immunotherapy to neoadjuvant chemotherapy. In addition, as previously reported, neither baseline TILs nor TMB nor other biomarkers may help in the identification of responders. Since the number of indications and TNBC patients receiving neoadjuvant ICIs is supposed to further increase soon, the identification of predictive biomarkers of response remains of pivotal importance. Not only PD-L1, TMB, TILs, but also novel focuses of research are under development, including gut microbiome, as witnessed by the publication of several recent studies on this topic in a wide range of tumor types [21,22,23,24].

Soon, further translational research is necessary to optimize the use of ICIs in early TNBC, and combinations of several biomarkers have the potential to be more impactful compared to a unique predictor of response, which has already shown important limitations. In addition, the current scenario of the adjuvant phase of treatment is becoming more complex considering some post-neoadjuvant trials, including CREATE-X and OlympiA, and immunotherapy has the potential to play a role also in this specific setting [25,26]. Several questions remain to be addressed, including the identification of TNBC patients that may benefit from the addition of ICIs as well as those that do not need neoadjuvant immunotherapy.

## Data Availability

Not applicable.

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
