# Peer review of "Current and Future Role of Neoadjuvant Chemoimmunotherapy for Early Triple-Negative Breast Cancer: Which Way to Go Forward"

_medicina, 2022, doi:10.3390/medicina58050600_

Round 1

Reviewer 1 Report

The editorial is clear, comprehensive and relevant, although some inconsistencies of the introduction (parts about mTNBC) are to revise.

Line 26: please review name of LAG-3

Line 28-30: Please review given ORR, considering distinguishing between therapeutic lines and citing Keynote trials properly.

Line 32: Citation 9 refers to a review including combination therapy. Maybe NCT01375842 was intended to be cited instead? 

Line 34: I suggest specifying PD-L1 overexpression according to atezolizumab label (IC score).

Author Response

Dear Reviewer,

Thank you so much for the time spent revising our paper. 

We have modified all the parts you indicated, as suggested.

All our changes have been highlighted in blue color in the revised paper.

Thank you again

Reviewer 2 Report

In this commentary the authors summarize and give their personal perspective about the results of 3 chemo-immunotherapy trials in triple negative breast cancer neo-adjuvant/adjuvant setting.

The article is well written but should undergo some minor revision.

In order to consistently report the findings of the studies I would suggest to:

  • Provide the pCR rates in the experimental and control arm of the Keynote-522 trial;
  • Provide information about the adverse events registered in the Keynote522 and GeparNuevo trials.

Additionally, in the conclusive section (What the future will be like? Open questions) authors should comment about the role of chemo-immunotherapy regimens in triple negative BRCA mutated breast cancer patients, given the recent approval of olaparib in the adjuvant setting. A comment about the role of post-neoadjuvant capecitabine (CREATE-X trial) in current scenario should also be added.

Author Response

Dear Reviewer,

Thank you so much for your time spent revising our paper. 

We modified several parts of the manuscript, as suggested.

All our changes related to your comments have been reported in green color.

Thank you again.

Regards

Round 2

Reviewer 1 Report

Citation Nr. 8 still refers to a HNSCC-trial (KeyNote-048).